 PLOS | ONE

# Prevalence and effects of gastro-oesophageal reflux during spirometry in subjects undergoing reflux assessment

Jerry Zhou[1]*, Ming Teo[2], Vincent Ho[1], John D. Brannan[3]

**1** School of Medicine, Western Sydney University, Campbelltown, NSW, Australia, **2** Department of Respiratory Medicine, Blacktown Hospital, Blacktown, NSW, Australia, **3** Department of Respiratory & Sleep Medicine, John Hunter Hospital, New Lambton, NSW, Australia

* j.zhou@westernsydney.edu.au

**Data Availability Statement:** All relevant data are within the paper and its Supporting Information files.

**Funding:** The author(s) received no specific funding for this work.

## Abstract

Variability during spirometry can persist despite control of technical and personal factors. We postulate spirometry induces gastro-oesophageal reflux (GOR), which may cause variability and affect results of spirometry. Fifty-eight (58) subjects undergoing GOR investigation with oesophageal manometry and 24hr pH monitoring were recruited. Oesophageal dysmotility and GOR were assessed as part of clinical care. Subjects performed 2 sets of spirometry separated by a 10-minute rest period. The assessment of GOR during spirometry procedure (defined by a lower oesophageal pH<4) started from the first set of spirometry and concluded when the second set of spirometry was completed. We calculated variability (%) of $FEV_1$, FVC and PEFR within each set as well as changes over 10-minutes. Twenty-six subjects (45%) recorded GOR during assessment. Of these, 23 subjects recorded GOR during the 10-minute rest period. Four subjects had GOR recorded only during spirometry tests. We did not find variability of spirometry parameters between the groups with and without GOR during spirometry procedure. However, in subjects with GOR, we found small but significant reductions of PEFR (0.5L/s, 8%, p<0.001) and $FEV_1$ (84 mL, 3%, p = 0.048) in the second set of spirometry compared to the first spirometry set. This pilot study demonstrates that GOR can occur during and following spirometry. Presence of GOR during spirometry in this patient population caused small decreases in PEFR and $FEV_1$ when it is repeated 10-minutes later however not increase variability in a single series of measurements.

## Introduction

Spirometry is the most widely used and accessible lung function test in respiratory medicine. Adequate repeatability is required to ensure high confidence in lung function interpretation. Thus, spirometry guidelines identify technical sources of variation during testing [1]. However, when the patient is employing an appropriate technique to perform the test there may be legitimate biological sources of variation in response to spirometry in persons with established lung disease. For example, patients with active asthma may have bronchoconstriction due to

**Competing interests:** The authors have declared that no competing interests exist.

the forced manoeuvre [2]. Other sources of pathophysiological variation have yet to be identified. In addition to individual test variation, some patients may show decreases in spirometry parameters 10-minutes following a bronchodilator assessment that is possibly independent of the effects of a standard dose of beta2-agonist. One possible pathophysiological event could be acid reflux induced by spirometry which could cause reflex bronchoconstriction [3] or upper airway acidification [4]. It may be possible upper airway acidification to reflux may impact upper airway function and negatively impact spirometry performance [5]. Lavorini et al., [6] observed cough-like expiratory efforts ("deflation cough") during spirometry manoeuvres, which the researchers correlated to gastro-oesophageal reflux (GOR). The frequency of deflation cough during spirometry was reduced after antacid administration.

There is a strong association between GOR disease and pulmonary diseases, including chronic obstructive pulmonary disease [7] and asthma [8], but the underlying mechanisms are unclear. It is suggested that hyperinflation, vigorous cough, or bronchospasm may increase intra-abdominal pressure and decrease the diaphragmatic contribution to sphincter tone, thereby promoting reflux of gastric contents [9–13]. Therefore, vigorous breathing manoeuvres performed as part of spirometry, from inspiration to total lung capacity (TLC) followed by forced expiration, may challenge the integrity of the gastro-oesophageal junction.

We hypothesise a standard spirometry manoeuvre may induce GOR events in reflux susceptible patients. We performed a pilot study to understand if GOR events associated with spirometry manoeuvres in persons with reflux symptoms referred for high-resolution oesophageal manometry and ambulatory 24-hour oesophageal pH monitoring. Secondary aims were to investigate if GOR occurring with spirometry affects spirometry results and variability.

## Materials and methods

### Recruitment

From July 2016 –October 2017, patients attended the GI Motility Clinic, Camden Hospital, NSW Australia, for suspected GOR evaluation. All patients underwent routine oesophagus function assessment (high-resolution oesophageal manometry) and 24 hr pH monitoring study, described below. Potential participants that met the inclusion criteria (adults with no history of major oesophageal motility disorders e.g. achalasia, oesophageal spasm) were approached and informed of the study. The 58 participants that provided informed written consent took part in a spirometry procedure, described in detail below. All procedures were in accordance with the Helsinki Declaration, and the study was approved by the local institutional review board: South Western Sydney Local Health District Human Ethics committee (Ref: HREC/15/LPOOL/462). The participants in this study are representative of a population that experiences frequent GOR symptoms and/or are diagnosed with GOR.

### High-resolution oesophageal manometry and 24-hour oesophageal pH monitoring

Patients referred to the Gastrointestinal Motility Clinic, Camden Hospital with reflux symptoms assessed by high-resolution oesophageal manometry (ManoScan ESO System, Medtronic, Minneapolis, MN, USA) and ambulatory 24-hour pH monitoring (Digitrapper pH-Z testing system, Medtronic, Minneapolis, MN, USA) as per standard protocols [14, 15].

We assessed oesophageal motility physiology in accordance to Chicago Classification Version 3.0 criteria [16]. Following manometry, the patients were intubated through the nose with a pH catheter adjusted to position the pH sensor 5 cm above the lower oesophageal sphincter. Patients went about their usual activities over a 24-hour period with the pH probe *in-situ* and

GOR during this period was quantified by the DeMeester score. GORD was diagnosed based on the 24-hour DeMeester score of $\geq 14.72$ [17] from the acquired recordings, (AccuView Software, Version 5.2 Medtronic, Minneapolis, MN, USA).

### Lower oesophageal pH monitoring during the assessment period of spirometry

A gastroenterologist (JZ) trained in spirometry enrolled patients into the study at the end of their 24-hour pH monitoring with the pH catheter still intubated. Subjects performed two sets of spirometry manoeuvres separated by a 10-minute rest period which mimicked the protocol for a pre and post bronchodilator spirometry test where two separate sets of spirometry are required. The assessment period for GOR started with the first set of spirometry and ended when the second set of spirometry was complete. The inspiratory and expiratory components of spirometry attempts were marked using the software, including the rest period on the pH recording device. The pH catheter was removed upon the completion of the second set of spirometry.

GOR is defined when lower oesophagus pH falls below 4 [17]. Two study investigators (JZ, VH) independently reviewed the oesophageal pH from the assessment period to determine the presence of GOR. They also assessed whether GOR occurred during the inspiratory phase or the forced expiratory phase of spirometry. Differences in their assessment by consensus discussion.

Acid reflux was quantified by the number of reflux episodes during each set of spirometry and during the rest period using AccuView software, as well as the number of reflux events and time-in-reflux during spirometry sets and rest period. A reflux event is denoted by the period between the pH below 4 and the pH returning to >4. Time-in-reflux is the average time of reflux events over the assessment period.

### Spirometry

Spirometry was conducted (QRS SpiroCard & Office Medic Software Version 4.4 Vectracor, Totowa, NJ, USA) via a laptop computer in accordance with the American Thoracic Society/ European Respiratory Society spirometry standard [18] to accommodate for the presence of the pH catheter within the nose. We made a minor modification to not use a nose clip due to discomfort from the intubated pH catheter. Instead participants were asked to pinch their nose closed. Prior to conducting the first set of spirometry, the study investigator checked that the patient was not experiencing reflux immediately prior to the manoeuvre. Subjects were instructed to inhale to total lung capacity before performing a forced expiratory manoeuvre and expiring to residual volume, for a minimum of three attempts and no more than five. We recorded the forced vital capacity (FVC), forced expiratory volume in 1 second ($FEV_1$) and peak expiratory flow rate (PEFR).

Subjects then had a 10-minute rest period, before performing a second set of spirometry (Fig 1).

### Data analysis

Statistical analysis was performed (SPSS 24.0 for Windows, SPSS Inc, Chicago, IL, USA). Data distribution was determined with Shapiro-Wilk normality test; mean and standard deviation calculated for normally distributed data, while median and interquartile range calculated for non-normally distributed data. Data groups were compared using the Student's t-test or Wilcoxon rank-sum test as appropriate for the distribution. Chi-square test was used to compare nominal variables between groups. Percentage of time-in-reflux was calculated as the total duration of all reflux events/duration of particular time period multiplied by 100. Variability within a set of spirometry was calculated for $FEV_1$, FVC and PEFR using the formula:

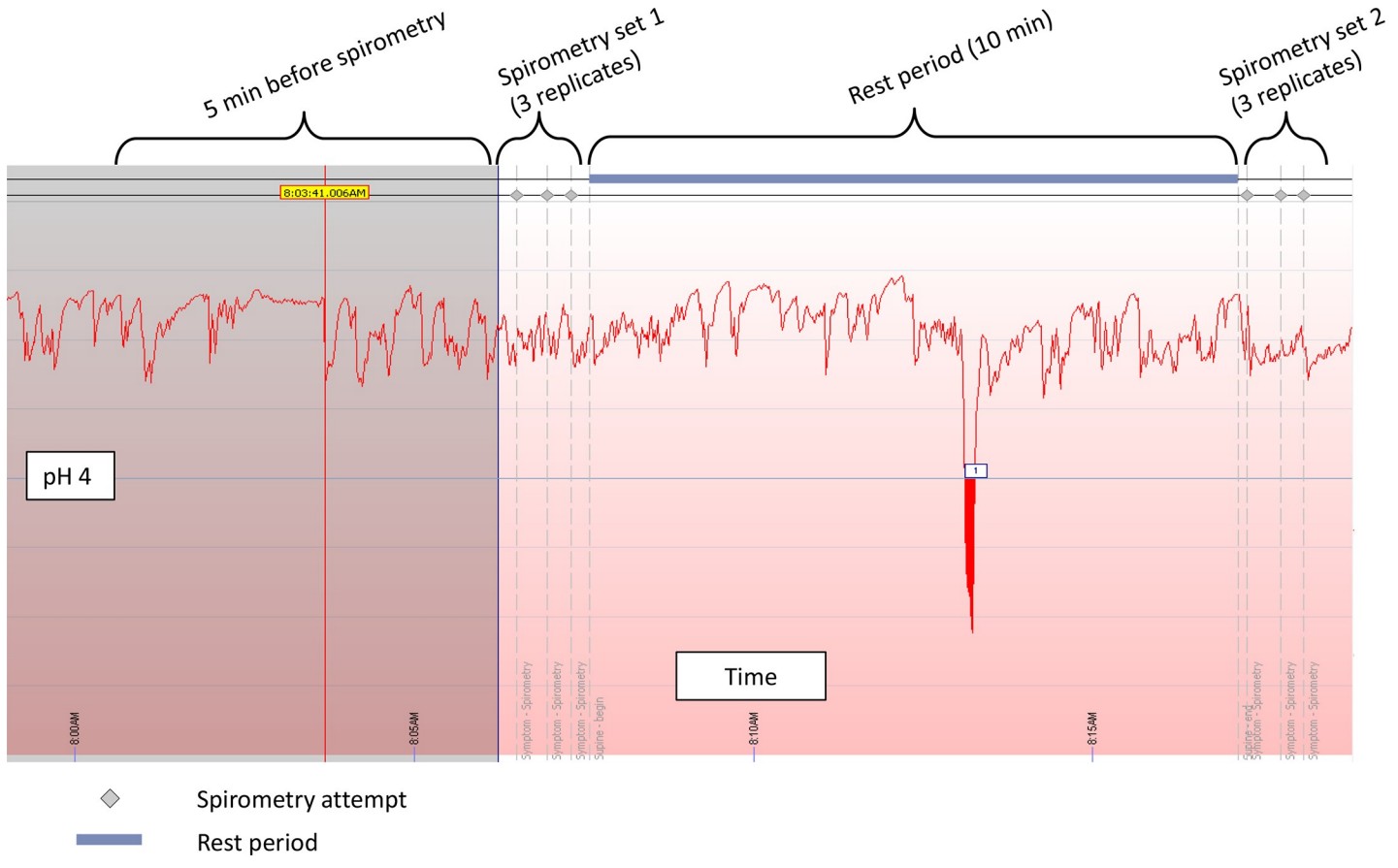

**Fig 1. Example of pH-monitoring during spirometry study.** Acid reflux events (pH <4.0) and duration are recorded by device software. Spirometry attempts are signified by grey diamond markers (top) which represent the beginning of a forced exhalation manoeuvre. Ten minute rest period is indicated with blue bar. Participants are in a seated position during the entire period of testing.

Variability = (max value–min value)/max value x 100 (expressed as a % value). Data is expressed as mean±SD unless otherwise indicated.

## Results

### Subject demographic and clinical investigation results

We enrolled 58 subjects (mean age 52yrs; range 21–79; 24 (41%) males, 34 (59%) females), with reflux symptoms, consecutively at the end of their 24-hour pH monitoring and all subjects completed the required 2 sets of spirometry. High-resolution oesophageal manometry assessment identified, 48 subjects (83%) with abnormalities predisposing to reflux (hiatus hernia and/or reduced lower oesophageal sphincter resting tone)while 10 (17%) had normal oesophageal function. Based on a cut-off DeMeester Score of 14.72, 39 (71%) subjects had clinically significant reflux on the preceding 24-hour oesophageal pH monitoring. The median DeMeester Score in these patients with significant reflux was 45.3 (IQR: 21.8 to 57.4) [19].

### GOR during spirometry assessment

Twenty-six (45%) subjects recorded GOR during spirometry assessment, and were compared with the 32 subjects that did not record GOR in Table 1. A greater proportion of subjects with GOR during spirometry assessment had reflux pathophysiology (89% vs 78%). Subjects with

**Table 1. Subject demographic data and clinical information from 24hr GORD assessment (n = 58).**

| | | Spirometry assessment | | Statistics |
|---|---|---|---|---|
| | | GOR present (n = 26) | GOR absent (n = 32) | p-values |
| **Gender** | | | | |
| | Male | 11 (42%) | 13 (41%) | 0.90 ($\chi^2$) |
| | Female | 15 (58%) | 19 (59%) | |
| **Age (mean ± SD) years** | | 49 ± 16 | 55 ± 14 | 0.15 (t-test) |
| **24 hr pH study** | | | | |
| | Diagnosed GORD | 20 (77%) | 19 (59%) | 0.27 ($\chi^2$) |
| | No GORD diagnosis | 6 (23%) | 13 (41%) | |
| | DeMeester (median ± IQR)[a] | 27 ± 35.9 | 21 ± 28.5 | 0.06 (Wilcoxon) |
| **Oesophageal manometry** | | | | |
| | Hiatus hernia present | 23 (88%) | 22 (69%) | 0.07 ($\chi^2$) |
| | Average resting LOS pressure (median ± IQR) mmHg | 14 ± 14.7 | 20 ± 17.3 | 0.06 (Wilcoxon) |
| | Reflux pathophysiology[b] | 23 (89%) | 25 (78%) | 0.01 ($\chi^2$) |
| | Normal physiology | 3 (11%) | 7 (22%) | |

[a]Gastro-oesophageal reflux disease (GORD) diagnosis based on DeMeester score ≥14.72 on 24-hour oesophageal pH monitoring; LOS: lower oesophageal sphincter
[b]Reflux pathophysiology refers to the presence of a hiatus hernia or low resting LOS pressure <13 mmHg); ns: non-significant (p >0.05).

GOR during spirometry were also more likely to be diagnosed with GORD (77% vs 59%; P = 0.27) and receive a higher DeMeester Score (mean 39.8±36.3 vs mean 26.4±24.8; P = 0.06) on the preceding 24-hour oesophageal pH monitoring, however these differences only approached statistically significance.

Of the 26 subjects who recorded GOR during spirometry assessment, GOR was most frequently observed during the 10-minute rest period (23/26, 88%; Table 2). A smaller proportion of subjects (17/26, 65%) had GOR during spirometry manoeuvres. A minority of subjects, had GOR exclusively during the first (1/25, 4%) and second sets (3/26, 12%) of spirometry.

The 17 subjects recorded GOR during spirometry manoeuvre had a combined total of 46 GOR events (distinct inverse peaks where pH<4) during testing, of which, 32 (70%) coincided with the inspiratory and expiratory phases of spirometry. Examples of GOR events during inspiration and expiration manoeuvres are shown in Fig 2A and 2B.

We found the highest mean number of GOR events of 3.2±1.9 in the 1st set of spirometry, compared to 2.9±2.9 and 2.3±2.4 during the rest period and second set of spirometry, however these differences were not significantly different (P = 0.24).

## Spirometry results and variability

The mean lung function values in this population were FVC 3.39L±1.0 and $FEV_1$ 2.72L±0.9. Comparisons of spirometry variables between 1st and 2nd sets are presented in Table 3. Subjects with observed GOR during spirometry assessment had a significantly lower mean PEFR

**Table 2. Analysis of GOR present during spirometry assessment (n = 26).**

| Time period | GOR present (%)[a] | Reflux events (median ± IQR) | Time in reflux (median ± IQR) % |
|---|---|---|---|
| 1st set spirometry | 6 (23%) | 2.5 ± 3.5 | 21.3 ± 39.7 |
| 10-min rest | 23 (88%) | 2.0 ± 3.0 | 3.6 ± 18.1 |
| 2nd set spirometry | 11 (42%) | 1.0 ± 2.0 | 5.8 ± 25.1 |

[a]Total in this column exceeds 26 as some subjects have GOR in 2 or more time periods.
[b]Reflux index: no. of GOR events divided by number of spirometry attempts.

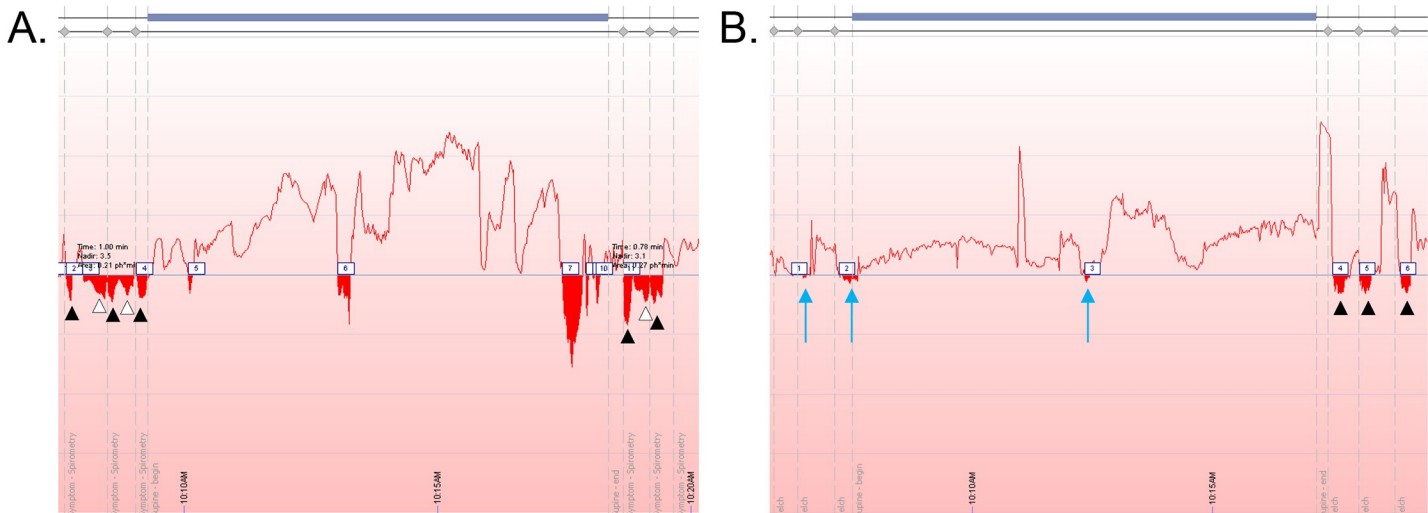

**Fig 2.** A) Examples of GOR occurring throughout assessment period. Black arrowheads indicate GOR events during forced expiratory manoeuvres while white arrowheads indicate GOR events during inspiration to Total Lung Capacity. B) blue arrows indicate subtle GOR events.

in the second spirometry compared to the first spirometry (0.5±0.6L/s lower, P<0.001). Although, the mean $FEV_1$ was also reduced in the second set of spirometry (84mL±0.2 lower, P = 0.048) the difference in volume would not appear to be of clinical significance. Mean FVC did not change between the two sets of spirometry.

In the group of participants without GOR during spirometry assessment, there was no significant difference in mean FEV1, FVC and PEFR between the two sets of spirometry.

The variability of FVC, $FEV_1$, and PEFR within each sets of spirometry is comparable between GOR present and GOR absent groups (Table 4). There is a slight reduction in variability within the second set compared to the first set for all FVC, $FEV_1$ and PEFR values but the differences were not statistically significant (P<0.05).

## Discussion

This pilot study demonstrates that GOR events occur during or following spirometry in almost half of patients presenting for assessment of GORD using 24-hour oesophageal pH monitoring. These findings support our primary hypothesis that spirometry can induce GOR.

**Table 3. Analysis of spirometry results (mean ± SD).**

| GOR present during spirometry assessment (n = 25)[a] | | | | |
|---|---|---|---|---|
| | **1st set** | **2nd set** | **Difference** | **Paired T-test** |
| **FVC (L)** | 3.41 ± 0.97 | 3.36 ± 0.94 | 0.048 ± 0.20 | 0.25 |
| **FEV1 (L)** | 2.77 ± 0.87 | 2.68 ± 0.85 | 0.084 ± 0.20 | 0.048[b] |
| **PEFR (L/s)** | 6.64 ± 2.18 | 6.14 ± 2.16 | 0.50 ± 0.60 | <0.001[b] |
| No GOR present during spirometry assessment (n = 32) | | | | |
| | **1st set** | **2nd set** | **Difference** | **Paired T-test** |
| **FVC (L)** | 3.35 ± 0.99 | 3.44 ± 1.11 | -0.092 ± 0.27 | 0.06 |
| **FEV1 (L)** | 2.68 ± 0.90 | 2.76 ± 0.99 | -0.082 ± 0.23 | 0.06 |
| **PEFR (L/s)** | 6.61 ± 2.20 | 6.47 ± 2.27 | 0.14 ± 0.72 | 0.30 |

[a]One spirometry study was omitted due to technical fault

[b]statistical significance p value <0.05

**Table 4. Comparison of parameter variability[a] (median% ± IQR) within a set of spirometry.**

| | Spirometry assessment | | | |
|---|---|---|---|---|
| | GOR present (n = 26) | | GOR absent (n = 32) | |
| Variability | 1st set | 2nd set | 1st set | 2nd set |
| FVC | 4.3 ± 7.4 | 4.0 ± 5.4 | 4.2 ± 4.9 | 4.2 ± 3.6 |
| FEV1 | 5.4 ± 7.4 | 5.4 ± 9.6 | 5.9 ± 5.1 | 4.5 ± 8.7 |
| PEFR | 13.0 ± 14.1 | 11.4 ± 18.1 | 13.3 ± 15.2 | 8.6 ± 13.1 |

[a]Variability within a set of spirometry was calculated for $FEV_1$, FVC and PEFR using the formula: Variability = (max value–min value)/max value x 100 (expressed as a % value)

Our study recruited a clinical population symptomatic of GOR undergoing high-resolution oesophageal manometry and 24-hour oesophageal pH monitoring. This was appropriate for a pilot study of this nature as we wanted to see if we were able to detect the physiological changes to support the primary hypothesis.

We found that GOR is associated with both inspiratory (inhaling to TLC from functional residual capacity during tidal breathing) and expiratory phases of spirometry. This has been observed in a similar study where the occurrence of deflation cough during maximal expiratory spirometry manoeuvres were associated with clinical features of GOR [6]. A potential mechanism for this, is the reduced intra-thoracic pressure during inspiration acts as a suctioning force that could draw gastric contents into the oesophagus. The diaphragmatic contraction during inspiration would result in increased lower oesophageal sphincter tone and would normally prevent GOR. However, in subjects with a hiatus hernia, this diaphragmatic squeeze acting on the hiatus hernia may propel contents within the hernia back into the oesophagus. Considering our initial findings, we propose that during forced expiration, the increase in intra-abdominal pressure may exceed intra-thoracic pressure, thus this pressure differential could cause movement of gastric contents into the oesophagus. Further, GOR may also be facilitated by diaphragmatic contraction being reduced during expiration. The reduced diaphragmatic tone in turn may reduce overall gastro-oesophageal junction tone, enabling retrograde movement of gastric contents. Thus, the observed strong association between GOR and forced expiration could be due to the combination of increased pressure gradients and reduced gastro-oesophageal tone during expiration manoeuvre.

A novel finding in this study was that the majority of GOR occurred during the 10-minute rest period. We postulate that the mechanical stresses associated with spirometry could temporarily weaken gastro-oesophageal junction tone leading to increased reflux susceptibility for a period of time after the first set of spirometry manoeuvres. This disruption of the junction integrity is likely to be transient as a low mean percentage time in reflux was recorded during the rest period and fewer GOR events were recorded in the second set of spirometry.

In the group of patients with reflux during spirometry assessment, we found a slightly lower PEFR in the second set of spirometry. Although a lower PEFR is commonly due to poorer patient effort, a possible explanation is reduction of large airway calibre, for instance from bronchoconstriction. Acid-induced bronchoconstriction caused by neurogenic inflammation as a result of tachykinin (substance P/neurokinin A) release upon acid stimulation has been demonstrated [20, 21]. The proposed mechanisms of this involve proton-activation of central vagal reflex and subsequent oesophageal-airway local axon reflex or microaspirate (reflux) stimulation of capsaicin-sensitive sensory neurons [22]. The possibility of reflux-induced bronchoconstriction having an effect on spirometry is supported further by our observation of the small but statistically significant reduction of FEV1. However these changes

were small and their clinical significance is debatable. Changes in PEF and FEV1 could also be due to changes in upper airway function during a forced manoeuvre [23]. There was no change in the variability of $FEV_1$, FVC or PEFR within the second spirometry set, and therefore we did not have evidence to suggest that poorer spirometry effort was the reason for the lower PEFR and $FEV_1$. The small changes in spirometry parameters may be attributed to participants in this study cohort having normal lung function and no history for respiratory diseases. Future study within a population with respiratory disease may lead to greater variability in lung function in the presence of acid reflux during spirometry.

Our findings are consistent with previous studies that observed indirect induction of reflex bronchoconstriction by gastric content that reaches the upper airways [10, 24] via tracheal irritant receptors believed to be situated in the upper airway epithelium [10]. A number of animal studies have demonstrated the activation of these receptors is associated with a vagally mediated reflex bronchoconstriction [25]. Exposures to acidic aerosols have also been shown to stimulate reflex bronchoconstriction [26, 27], most likely by stimulation of irritant receptors in the tracheobronchial tree.

Acute upper airway acidification has also been shown to cause changes in upper airway calibre representing upper airway dysfunction, for example variability vocal cord abduction or increased adduction [3, 26, 27]. A limitation of our pH measurement was limited to the mid-distal oesophagus, and we could not measure acid reflux in the upper oesophagus, pharynx or larynx. Therefore we cannot confirm a possible cause for upper airway dysfunction if this were responsible, as knowing this would be able to reveal if these changes could be due to upper airway reflux. Further studies should investigate this mechanism in more detail by identify changes in upper airway pH with spirometry in both susceptible subjects who demonstrate decreases in post bronchodilator spirometry and including the effects in a well characterised healthy population.

## Conclusion

This study demonstrates that GOR events do occur during spirometry in a group of patients who are predisposed to GOR or have significant GOR disease. Further, a high prevalence of GOR occurred during a 10-minute rest period between two sets of spirometry. The observation of reductions in some spirometry parameters 10-minutes after an initial set of spirometry manoeuvres suggests reductions in airway calibre may be associated with spirometry-induced GOR or the presence of GORD. Further research investigating this relationship may determine if these reductions in spirometry parameters are reflective of the predisposition to GOR events and increases in airway acidity.

## Supporting information

**S1 File. Study data.** Summary of participant characteristics, oeosphagogastric findings, 24hr pH study results, and spirometry evaluation.
(TXT)

## Author Contributions

**Conceptualization:** Jerry Zhou, John D. Brannan.

**Data curation:** Jerry Zhou, Ming Teo, Vincent Ho, John D. Brannan.

**Formal analysis:** Jerry Zhou, John D. Brannan.

**Investigation:** Jerry Zhou.

**Methodology:** Jerry Zhou, Ming Teo, Vincent Ho, John D. Brannan.

**Project administration:** Jerry Zhou.

**Resources:** Jerry Zhou, John D. Brannan.

**Software:** Jerry Zhou.

**Validation:** Jerry Zhou, Vincent Ho, John D. Brannan.

**Visualization:** Jerry Zhou.

**Writing – original draft:** Jerry Zhou, John D. Brannan.

**Writing – review & editing:** Jerry Zhou, Ming Teo, Vincent Ho, John D. Brannan.

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
