## [Decision Letter · Decision Letter 0]

27 Nov 2019

PONE-D-19-25405

Prevalence and effects of gastro-oesophageal reflux during spirometry in subjects undergoing reflux assessment

PLOS ONE

Dear Dr. Jerry Zhou,

Thank you for submitting your manuscript to PLOS ONE. After careful consideration, we feel that it has merit but does not fully meet PLOS ONE’s publication criteria as it currently stands. Therefore, we invite you to submit a revised version of the manuscript that addresses the points raised during the review process.

The manuscript is interesting but it could be considered for publication after appropriate revision.

We would appreciate receiving your revised manuscript by 25th of January 2020. To enhance the reproducibility of your results, we recommend that if applicable you deposit your laboratory protocols in protocols.io, where a protocol can be assigned its own identifier (DOI) such that it can be cited independently in the future. For instructions see: http://journals.plos.org/plosone/s/submission-guidelines#loc-laboratory-protocols

We look forward to receiving your revised manuscript.

Kind regards,

Fabio Luigi Massimo Ricciardolo

Academic Editor

PLOS ONE

Journal Requirements:

2. In the ethics statement in the Methods and online submission information, please ensure that you have specified whether consent was informed.

3. In your Methods section, please provide additional information about the participant recruitment method and the demographic details of your participants. Please ensure you have provided sufficient details to replicate the analyses such as: a) the recruitment date range (month and year), b) a description of any inclusion/exclusion criteria that were applied to participant recruitment, c) a statement as to whether your sample can be considered representative of a larger population and e) a description of how participants were recruited

Additional Editor Comments (if provided):

This is an interesting manuscript and the authors should follow the reviewer' suggestions in order to improve the impact of the manuscript.

Reviewers' comments:

Reviewer's Responses to Questions

**Comments to the Author**

1. Is the manuscript technically sound, and do the data support the conclusions?

Reviewer #1: Yes

Reviewer #2: Yes

2. Has the statistical analysis been performed appropriately and rigorously? 

Reviewer #1: Yes

Reviewer #2: Yes

3. Have the authors made all data underlying the findings in their manuscript fully available?

Reviewer #1: Yes

Reviewer #2: Yes

4. Is the manuscript presented in an intelligible fashion and written in standard English?

Reviewer #1: Yes

Reviewer #2: Yes

5. Review Comments to the Author

Reviewer #1: This study demonstrates GOR may result in very mild physiologic changes but not clinically significant changes in the patients studied. The authors imply specific changes despite not proven by statistics.

Reviewer #2: The present work was aimed to show the occurrence of gastro-oesophageal reflux (GOR) during or soon after spirometry (i.e.: forced expiratory maneuver following full inspiration to total lung capacity) and to assess the potential consequences of in-expiratory effort-related GOR in terms of subsequent respiratory functional results and their variability.

The rational is that in individuals who potentially may have gastro-oesophageal reflux disease (GORD), as suggested by peculiar symptoms or anatomic-functional predisposing conditions, the increased positive pressure gradient occurring during in-expiratory efforts between abdomen and thorax may induce distal or even proximal acid reflux causing reflex bronchoconstriction or upper airway stimulation with subsequent increase in airflow resistance.

In this cohort of subjects with high probability of having GORD, the Authors found that GOR was present during or following spirometry in almost half of the subjects (26/58). In contrast with subjects without GOR during or following spirometry, those with GOR showed in the subsequent set of spirometry a significant reduction of some functional parameters, suggesting the occurrence of the GOR-related airflow obstruction.

In addition, the results show that in 39 subjects over 58 suspected of GORD who were diagnosed as having GORD by standard criteria based on 24-hours oesophageal pH monitoring, about 50% (20/39) exhibited spirometry-induced GOR and the resting 50% (19/39) not. Conversely, in 19 subjects who did not have GORD, 6/19 (about one/third) exhibited spirometry-induced GOR and 13/19 not. Thus, repeated spirometry seems not useful to increase the diagnosis of GORD.

General comments

The presence of GOR during spirometry is not a novelty and in fact deflation-related peak expiratory flows occurring near residual volume at the end of maximal expiratory maneuvers has been suggested as a marker of GOR (see Lavorini et al. Chest 2011;140:690).

The new findings of this work are the more frequent occurrence of GOR throughout 10 minutes after spirometry than during spirometry and the mild reduction of some functional indices (such as PEF and FEV1) in the subsequent set of spirometry.

Surprisingly GOR was recorded also during slow inspiration toward TLC before the maximal forced expiration.

The methods to assess GOR are up to date.

The numbers in the tables 1 and 2 are difficult to follow and sometimes do not fit with the text. Please check it.

The statistics are adequate.

Minor comments

I do not think these findings may explain the paradoxical increase of airflow obstruction after a bronchodilator test performed to assess the reversibility of baseline airway obstruction and I will skip this sentence from the discussion.

Some English mistakes here and there need to be amended.

6. PLOS authors have the option to publish the peer review history of their article (what does this mean?). If published, this will include your full peer review and any attached files.

Reviewer #1: No

Reviewer #2: No

---

## [Author Response · Author response to Decision Letter 0]

7 Dec 2019

Thank you for reviewing and providing feedback for our manuscript: 

Manuscript ID: [PONE – D-19-25405] – [EMID:d5e0088e3ed7c23b]

Title: Prevalence and effects of gastro-oesophageal reflux during spirometry in subjects undergoing reflux assessment

Authors: Jerry Zhou, Ming Teo, Vincent Ho, John D. Brannan

In Response to Journal Requirements: 

1. The manuscript has been formatted in accordance with PLOS ONE style

2. Manuscript amended to address how consent was provided: “Participants were informed of the study and written consent was provided from those participating in the study.” 

3. Additional section added to Methods: Recruitment; detailing a) recruitment date range b) inclusion criteria c) representation of larger population e) recruitment process

4. Fig 1 and 2 uploaded and converted in PACE digital diagnostic tool

In Response to Reviewer #1

1. This study demonstrated the presence of acid reflux during and immediately following spirometry in individuals with reflux symptoms. This cohort of individuals who have GOR symptoms was selected to maximise the chance of reflux during spirometry. The presence of reflux was also noted to have a significance effect on FEV1 and PEFR (Table 2), although as the reviewer correctly states the differences are small and may not be of clinical significance. This may be due to the test group selection and may translate to larger variations in a test group with lung function disorders. The manuscript has been amended to address the small changes in FEV1 and PEFR, and added a section discussing the choice of test group.

In Response to Reviewer #2 

1. The study by Lavorini et al., has been acknowledged and the amended manuscript compares its results to this study.

2. The novelty of reflux immediately after spirometry has been emphasised in the amended manuscript

3. Tables 1 & 2 have been amended, data that were not utilised in the text and discussion are removed to improve readability. Data removed from Table 1: Hiatus hernia size, Presence of low resting pressure, minor motility disorder combined to reflux pathophysiology. Data removed from Table 2: GOR only in this period, Mean reflux index. Values in text have also been updated to match table values. 

4. “Paradoxical increase of airflow obstruction after a bronchodilator test performed to assess the reversibility of baseline airway obstruction” sentence is removed in amended manuscript. 

5. English mistakes amended.

---

## [Editor Report · Decision Letter 1]

7 Jan 2020

PONE-D-19-25405R1

Prevalence and effects of gastro-oesophageal reflux during spirometry in subjects undergoing reflux assessment

PLOS ONE

Dear Dr. Jerry Zhou,

Thank you for submitting your manuscript to PLOS ONE. After careful consideration, we feel that it has merit but does not fully meet PLOS ONE’s publication criteria as it currently stands. Therefore, we invite you to submit a revised version of the manuscript that addresses the points raised during the review process.

The authos should revise the manuscript following reviewers' suggestions.

In addition, I found old papers cited in the reference list concerning the mechanisms of association between GOR and bronchospasm/asthma and, in particular, I would add some sentences about the role of neurogenic inflammation as a main mechanism of bronchoconstriction due to airway acidification (protons-induced bronchoconstriction). I suggest to update the reference list.

We would appreciate receiving your revised manuscript by February 4th. To enhance the reproducibility of your results, we recommend that if applicable you deposit your laboratory protocols in protocols.io, where a protocol can be assigned its own identifier (DOI) such that it can be cited independently in the future. For instructions see: http://journals.plos.org/plosone/s/submission-guidelines#loc-laboratory-protocols

We look forward to receiving your revised manuscript.

Kind regards,

Fabio Luigi Massimo Ricciardolo

Academic Editor

PLOS ONE

Reviewers' comments:

Reviewer 1:

This study demonstrates GOR may result in very mild physiologic changes but not clinically significant changes in the patients studied. The authors imply specific changes despite not proven by statistics.

Reviewer 2:

The present work was aimed to show the occurrence of gastro-oesophageal reflux (GOR) during or soon after spirometry (i.e.: forced expiratory maneuver following full inspiration to total lung capacity) and to assess the potential consequences of in-expiratory effort-related GOR in terms of subsequent respiratory functional results and their variability.

The rational is that in individuals who potentially may have gastro-oesophageal reflux disease (GORD), as suggested by peculiar symptoms or anatomic-functional predisposing conditions, the  increased positive pressure gradient occurring during in-expiratory efforts between abdomen and thorax may induce distal or even proximal acid reflux causing reflex bronchoconstriction or upper airway stimulation with subsequent increase in airflow resistance.

In this cohort of subjects with high probability of having GORD, the Authors found that GOR was present during or following spirometry in almost half of the subjects (26/58). In contrast with subjects without GOR during or following spirometry, those with GOR showed in the subsequent set of spirometry a significant reduction of some functional parameters, suggesting the occurrence of the GOR-related airflow obstruction.

In addition, the results show that in 39 subjects over 58 suspected of GORD who were diagnosed as having GORD by standard criteria based on 24-hours oesophageal pH monitoring, about 50% (20/39) exhibited spirometry-induced GOR and the resting  50% (19/39) not. Conversely, in 19 subjects who did not have GORD, 6/19 (about one/third) exhibited spirometry-induced GOR and 13/19 not. Thus, repeated spirometry seems not useful to increase the diagnosis of GORD.

General comments

The presence of GOR during spirometry is not a novelty and in fact deflation-related peak expiratory flows occurring near residual volume at the end of maximal expiratory maneuvers has been suggested as a marker of GOR (see Lavorini et al. Chest 2011;140:690).

The new findings of this work are the more frequent occurrence of GOR throughout 10 minutes after spirometry than during spirometry and the mild reduction of some functional indices (such as PEF and FEV1) in the subsequent set of spirometry.

Surprisingly GOR was recorded also during slow inspiration toward TLC before the maximal forced expiration.

The methods to assess GOR are up to date.

The numbers in the tables 1 and 2 are difficult to follow and sometimes do not fit with the text. Please check it.

The statistics are adequate.

Minor comments

I do not think these findings may explain the paradoxical increase of airflow obstruction after a bronchodilator test performed to assess the reversibility of baseline airway obstruction and I will skip this sentence from the discussion.

Some English mistakes here and there need to be amended.

---

## [Author Response · Author response to Decision Letter 1]

7 Jan 2020

In Response to Journal Requirements: 

1. The manuscript has been formatted in accordance with PLOS ONE style

2. Manuscript amended to address how consent was provided: “Participants were informed of the study and written consent was provided from those participating in the study.” 

3. Additional section added to Methods: Recruitment; detailing a) recruitment date range b) inclusion criteria c) representation of larger population e) recruitment process

4. Fig 1 and 2 uploaded and converted in PACE digital diagnostic tool

5. Support file attached (study data) along with update in manuscript

6. The role of neurogenic inflammation as a mechanism for reflux-induced bronchoconstriction is a salient point and supports our findings that GOR affects spirometry tests. Amendment has been made to highlight its significance in the Discussion (Line 265-269) along with the proposed mechanisms of neurogenic inflammation in bronchoconstriction as a result of vagal reflex and microaspirate stimulation. 

In Response to Reviewer #1

1. This study demonstrated the presence of acid reflux during and immediately following spirometry in individuals with reflux symptoms. This cohort of individuals who have GOR symptoms was selected to maximise the chance of reflux during spirometry. The presence of reflux was also noted to have a significance effect on FEV1 and PEFR (Table 2), although as the reviewer correctly states the differences are small and may not be of clinical significance. This may be due to the test group selection and may translate to larger variations in a test group with lung function disorders. The manuscript has been amended to address the small changes in FEV1 and PEFR, and added a section discussing the choice of test group.

In Response to Reviewer #2 

1. The study by Lavorini et al., has been acknowledged and the amended manuscript compares its results to this study.

2. The novelty of reflux immediately after spirometry has been emphasised in the amended manuscript

3. Tables 1 & 2 have been amended, data that were not utilised in the text and discussion are removed to improve readability. Data removed from Table 1: Hiatus hernia size, Presence of low resting pressure, minor motility disorder combined to reflux pathophysiology. Data removed from Table 2: GOR only in this period, Mean reflux index. Values in text have also been updated to match table values. 

4. “Paradoxical increase of airflow obstruction after a bronchodilator test performed to assess the reversibility of baseline airway obstruction” sentence is removed in amended manuscript. 

5. English mistakes amended.

---

## [Decision Letter · Decision Letter 2]

4 Feb 2020

Prevalence and effects of gastro-oesophageal reflux during spirometry in subjects undergoing reflux assessment

PONE-D-19-25405R2

Dear Dr. Jerry Zhou,

We are pleased to inform you that your manuscript has been judged scientifically suitable for publication and will be formally accepted for publication once it complies with all outstanding technical requirements.

With kind regards,

Fabio Luigi Massimo Ricciardolo

Academic Editor

PLOS ONE

Additional Editor Comments (optional):

Reviewers' comments:

Reviewer's Responses to Questions

**Comments to the Author**

1. If the authors have adequately addressed your comments raised in a previous round of review and you feel that this manuscript is now acceptable for publication, you may indicate that here to bypass the “Comments to the Author” section, enter your conflict of interest statement in the “Confidential to Editor” section, and submit your "Accept" recommendation.

Reviewer #1: All comments have been addressed

Reviewer #2: All comments have been addressed

2. Is the manuscript technically sound, and do the data support the conclusions?

Reviewer #1: Yes

Reviewer #2: Yes

3. Has the statistical analysis been performed appropriately and rigorously? 

Reviewer #1: Yes

Reviewer #2: Yes

4. Have the authors made all data underlying the findings in their manuscript fully available?

Reviewer #1: Yes

Reviewer #2: Yes

5. Is the manuscript presented in an intelligible fashion and written in standard English?

Reviewer #1: Yes

Reviewer #2: Yes

6. Review Comments to the Author

Reviewer #1: Author responses are appropriate and ms acceptable for publication. The ms will serve as a guide to pulmonologists when interpreting pulmonary function studies in patients with GERD

Reviewer #2: General comments

The amended version of this work is definitively improved, more easily readable and better written than the original paper.

I do not have more questions or criticisms to rise.

The main practical advice following the results of this study is that in patients with baseline obstructive ventilatory defect, knowing to have GORD or predisposing conditions to GOR, the spirometric response to a subsequent test of bronchial responsiveness using short-acting bronchodilators (if required) should be assessed after an interval of time longer than 10 minutes (20 or better 30 min) to avoid spurious results.

Perhaps this suggestion could be added at the end of discussion section if Authors agree.

7. PLOS authors have the option to publish the peer review history of their article (what does this mean?). If published, this will include your full peer review and any attached files.

Reviewer #1: Yes: Arthur F Gelb MD

Reviewer #2: No

---

## [Editor Report · Acceptance letter]

7 Feb 2020

PONE-D-19-25405R2 

Prevalence and effects of gastro-oesophageal reflux during spirometry in subjects undergoing reflux assessment 

Dear Dr. Zhou:

I am pleased to inform you that your manuscript has been deemed suitable for publication in PLOS ONE. Congratulations! Your manuscript is now with our production department. 

With kind regards,

on behalf of

Professor Fabio Luigi Massimo Ricciardolo 

Academic Editor

PLOS ONE